# Carbon dioxide electroreduction to $C_2$ products over copper-cuprous oxide derived from electrosynthesized copper complex

Qinggong Zhu[1], Xiaofu Sun[1,2], Dexin Yang[1,2], Jun Ma[1], Xinchen Kang[1,2], Lirong Zheng[3], Jing Zhang[3], Zhonghua Wu [3] & Buxing Han[1,2,4]

Efficient electroreduction of carbon dioxide to multicarbon products in aqueous solution is of great importance and challenging. Unfortunately, the low efficiency of the production of $C_2$ products limits implementation at scale. Here, we report reduction of carbon dioxide to $C_2$ products (acetic acid and ethanol) over a 3D dendritic copper-cuprous oxide composite fabricated by in situ reduction of an electrodeposited copper complex. In potassium chloride aqueous electrolyte, the applied potential was as low as −0.4 V vs reversible hydrogen electrode, the overpotential is only 0.53 V (for acetic acid) and 0.48 V (for ethanol) with high $C_2$ Faradaic efficiency of 80% and a current density of 11.5 mA cm$^{-2}$. The outstanding performance of the electrode for producing the $C_2$ products results mainly from near zero contacting resistance between the electrocatalysts and copper substrate, abundant exposed active sites in the 3D dendritic structure and suitable copper(I)/copper(0) ratio of the electrocatalysts.

[1] Beijing National Laboratory for Molecular Sciences, CAS Key Laboratory of Colloid, Interface and Chemical Thermodynamics, CAS Research/Education Center for Excellence in Molecular Sciences, Institute of Chemistry, Chinese Academy of Sciences, Beijing 100190, P. R. China. [2] University of Chinese Academy of Sciences, Beijing 100049, P. R. China. [3] Institute of High Energy Physics, Chinese Academy of Sciences, Beijing 100049, P. R. China. [4] Shanghai Key Laboratory of Green Chemistry and Chemical Processes, School of Chemistry and Molecular Engineering, East China Normal University, Shanghai 200062, P. R. China. Correspondence and requests for materials should be addressed to J.M. (email: majun@iccas.ac.cn) or to B.H. (email: hanbx@iccas.ac.cn)

Carbon dioxide ($CO_2$) is an abundant $C_1$ resource[1]. Water, as a cheap resource, is a suitable electrochemical reaction medium and hydrogen source. In the quest for developing techniques for $CO_2$ utilization, electrochemical reduction offers a potential route to economical conversion of $CO_2$ and water into value-added chemicals and fuels[2]. Until now, extensive efforts have been devoted to the development of catalysts for electrocatalytic reduction of $CO_2$ in water[3–5]. Despite the notable progress, it is still highly desirable to develop efficient electrocatalysts to steer the reaction pathways toward higher value products and reduce reaction overpotential.

Products with two or more carbons ($C_{2+}$ product), such as acetic acid and ethanol, are very useful chemicals or fuels. Efficient electrochemical reduction of $CO_2$ to $C_{2+}$ products is of great importance[6–9]. Literature survey shows that Cu-based catalysts are the most efficient material for electrocatalytic reduction of $CO_2$, and Cu is a key component for forming multicarbon products[10–20]. On Cu-derived oxide electrocatalysts, $C_2$ or $C_{2+}$ products could be formed with Faradaic efficiency (FE) of ~70%[11,12]. Cu nanoparticles, Cu nanocubes and Cu nanoclusters show selectivity up to 60% for transformation of $CO_2$ to $C_{2+}$ products[9,13–15]. Plasma-activated Cu or Cu nanocube catalysts exhibit high FE to ethylene (60%) or ethylene/ethanol (45%/22%)[16,17]. Recent study indicates that the FE of ethylene on Boron-modified Cu-based electrocatalysts could reach 79%[6]. Cu gas diffusion electrode was found to be efficient for $CO_2$ reduction to ethylene with a FE of 70% at high current density[8]. Besides, $Cu_2S$–Cu core–shell–vacancy catalyst could boost $CO_2$ reduction to $C_2$ product with a FE of 32% at −0.95 V vs RHE[18]. Metal-free catalysts, such as nitrogen-doped mesoporous carbon, are also used as efficient catalysts for $CO_2$ reduction to ethanol (44%)[19]. It was also demonstrated that bimetallic catalysts have improved activity and selectivity for $CO_2$ reduction to hydrocarbons[20]. Comparing with production of $C_1$ chemicals or fuels, the reports on electrochemical reduction of $CO_2$ to $C_{2+}$ products is very limited. Especially, exploration of electrochemical systems to mediate multiple proton transfers with low overpotential is still a challenge for $CO_2$ reduction to $C_{2+}$ products because the C–C coupling is difficult. It is both scientifically and practically appealing to explore efficient electrolysis systems for $CO_2$ reduction to $C_{2+}$ product under heterogeneous catalysis conditions in aqueous media.

Designing robust electrocatalysts is very important for efficient $CO_2$ reduction[21,22]. Previous reports indicate that heterogeneous metal-based catalysts derived from molecular complexes can be favorable electrocatalysts[23,24]. The catalytic properties of the catalysts can be tailored by adjusting and regulating the composition of the complexes, and they exhibit excellent catalytic activity toward water splitting and oxygen reduction[25–29]. So far, it is also an approach toward the fabrication of electrocatalysts, such as additive controlled electrodeposited Cu catalysts for $CO_2$ reduction[30–32]. However, as a precursor, the complex was generally prepared ex situ or used as a homogeneous additive in decomposition of the new structure in $CO_2$ electroreduction.

Electrodeposition is a very useful technique that requires simple equipment with precise control of the growth processes, purity, structures, and morphologies of the deposits. The deposits are well suited for many applications, such as fuel cells, batteries, and sensors[33–35]. This method has also been used to prepare electrocatalysts for $CO_2$ reduction[36–39]. For example, electrodeposited In or Bi electrode can promote $CO_2$ reduction to CO with high selectivity[36,37]. $Cu_2O$ prepared by electrodeposition method can control the catalytic selectivity of $CO_2$ to hydrocarbons[11]. In situ deposited Cu nanodendrites as gas diffusion layer can reach high FE for $C_2$ products[38]. In general, it is easy to produce two-dimensional (2D) materials using the direct electrodeposition method. While an electrodeposition process can also create a 3D material, it is difficult to control the microstructure with desired features[33]. Therefore, it is very interesting to explore more practical methods for obtaining stable 3D structures with desirable structures for $CO_2$ electrocatalytic reduction.

The immobilization of molecular catalysts onto surfaces provides some obvious advantages over heterogeneous catalysts, such as controllable 3D structure, fast electron transfer rate, etc. Recently, an electrosynthesis method has been used to prepare metal organic complex films supported on conducting substrates[40,41]. This route constructs the metal organic complex films by assembling the reduced organic linkers and oxidized metal ions on a conductive substrate. The synthetic method is currently emerging as a promising approach for producing films of specific configurations, particularly in sensors and electric devices[40].

Herein, we report a more direct and facile method to electrodeposit a high-surface area Cu-complex film onto conductive substrates. A 3D dendritic Cu–$Cu_2O$ composite catalyst (denoted as Cu–$Cu_2O$/Cu) can be obtained via in situ growth and decomposition of the corresponding Cu-complex film. By using the Cu-complex electrodes as precursors and templates, the Cu–$Cu_2O$/Cu electrode has unique characteristics, such as dendritic 3D structure as well as near zero contact resistance between the catalyst and the substrate. The in situ synthesized electrode has outstanding catalytic performance for $CO_2$ reduction to $C_2$ products in KCl aqueous electrolyte. At −0.4 V vs RHE, the FE of $C_2$ products could reach 80% with a current density of 11.5 mA cm$^{-2}$. The respective overpotentials for acetic acid and ethanol are 0.53 and 0.48 V, respectively.

## Results

**Electrosynthesis and characterization of Cu-complex/Cu.** The electrodes were fabricated via two steps, including in situ electrodeposition of Cu-complex film on Cu substrate (denoted as Cu-complex/Cu) and in situ electroreduction of the Cu$^{II}$ in Cu-complex film to obtain Cu–$Cu_2O$/Cu electrode, which are discussed in the following.

In this work, six Cu-complexes with different ligands, which are denoted as Complex-1 to Complex-6 (Supplementary Scheme 1), were in situ deposited on Cu substrate by electrosynthesis method[40,41], where the numbers correspond to that of the ligands shown in Supplementary Scheme 1. The procedures to prepare the Cu-complex/Cu is shown schematically in Fig. 1a taking Complex-1 (pyromellic acid, 1,2,4,5-$H_4$BTC) as the example. Briefly, the electrosynthesis of Complex-1/Cu was carried out in a two-electrode system, which consisted mainly of Cu foil anode, Cu foil cathode, ligand, and electrolyte solution. The Cu$^{II}$ ions produced from electro-oxidation of the Cu anode (Cu substrate) coordinated with the electro-deprotonated organic ligands generated at the cathode, and then the complex film grew in situ on Cu substrate surface.

The growth of Complex-1 film on Cu substrate at different electrodeposition times were investigated using scanning electron microscopy (SEM, Fig. 1b). It is demonstrated that Complex-1 film with 3D hierarchical structure was formed with increasing electrodeposition time. The side-view SEM image of the Complex-1/Cu with electrodeposition time of 60 min is also given (Fig. 1b, Supplementary Figs. 1–6), indicating that the complex film of about 60 μm compacted tightly with the Cu substrate without any noticeable crack. This is understandable because the Cu ions were generated from the Cu substrate and the complexes grew on the substrate surface.

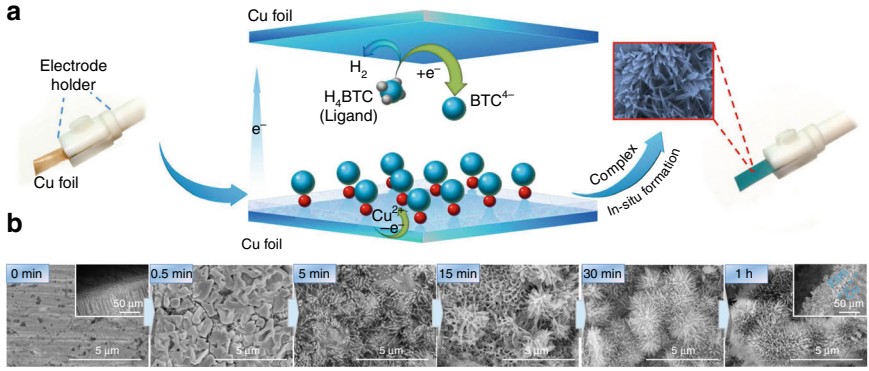

**Fig. 1** In situ anodic electrodeposition of copper complex on copper foil. **a** Schematic illustration of the process to prepare Cu-Complex-1 film on Cu substrate. H$_4$BTC = pyromellic acid, BTC$^{4-}$ = tetraanion of pyromellitic acid; **b** SEM images of Complex-1 formed on Cu substrate at electrodeposition times of 0, 0.5, 5, 15, 30, 60 min. The side-view of the film obtained at 60 min is also given in the figure as an insert

The complex with electrodeposition time of 60 min was further characterized by Fourier transform infrared (FT-IR) and X-ray diffraction (XRD) methods (Supplementary Figs. 7 and 8). The results show that the complexes were synthesized successfully and there was no evident impurity in the complexes. The X-ray photoelectron spectroscopy (XPS) spectra indicate that Cu$^{II}$ was the major metallic species in the complex (Supplementary Fig. 9). N$_2$ adsorption/desorption method was used to determine the surface areas and porous properties of the complex, and the results (Supplementary Fig. 10 and Supplementary Table 1) demonstrate that the complex has porous structure, and the molar ratio of Cu and ligand in the complex was 2.69, which was obtained from the Cu and C contents in the complexes determined by induced coupled plasma (ICP) methods (Supplementary Table 2) and the complex loading was 0.67 mg cm$^{-2}$ after 60 min deposition (Supplementary Table 3). Small angle X-ray scattering (SAXS) technique was employed to analyze the fractal structure. The surface fractal ($D_s$) values were obtained from the ln$I(h)$ vs ln($h$) plots (Supplementary Fig. 11). It can be seen that the surface of the complexes was coarse. The surface roughness factors of the complex film and Cu substrate were also determined by measuring the double layer capacitance using the reported method[4,42]. The results (Supplementary Fig. 12 and Supplementary Table 4) indicate that the surface of the complex film was much rougher than that of Cu substrate. All the results above confirm that the Cu-complex was successfully synthesized on the Cu substrate by the electrodeposition method. This is understandable because multibenzenecarboxylates are commonly used ligands in preparation of metal complexes[43–48]. The formation of the complexes was further proved by successful synthesis of the single crystal with 3D supramolecular structure (Supplementary Figs. 13−18 and Supplementary Tables 15−21) using interface growth method.

**Decomposition and characterization of Cu–Cu$_2$O/Cu.** We found that the Cu$^{II}$ in Cu-complex/Cu electrode was reduced quickly in N$_2$ or CO$_2$-saturated 0.1 M KCl aqueous electrolyte at applied potential of −0.4 V vs RHE. The current density at different electroreduction times are given in Fig. 2a. The current density was reduced to near zero after 5 min in N$_2$-saturated electrolyte. The current density in CO$_2$-saturated electrolyte became constant after 5 min, and the current after 5 min was originated from the CO$_2$ electroreduction, which will be discussed in detail in the following sections. The change of electrode properties with electroreduction times was studied by SEM, XRD, XPS, and X-ray absorption fine structure spectroscopy (XAFS) methods. SEM images (Fig. 2b and Supplementary Fig. 19) show

clearly that the 3D flower-like morphology of the complex was transformed into the dendritic 3D structure completely in initial 5 min of electroreduction. The morphology did not change notably after 5 min, and the thickness of the dendritic layer was about 40 μm. The XPS spectra of the catalyst film at different electroreduction times given in Fig. 2c, d indicate that the valence states of Cu changed from Cu$^{II}$ to Cu$^{I}$ and Cu$^0$ during the initial stage of electroreduction. The film after electroreduction was composed of Cu and Cu$_2$O as can be known from the XRD patterns (Supplementary Fig. 20). The ratio of Cu$^{I}$/Cu$^0$ became constant after 5 min of electroreduction (Supplementary Fig. 21 and Supplementary Table 5). The XAFS modeling and analysis results (Supplementary Figs. 22, 23 and Supplementary Table 6) further revealed the existence of Cu–O and Cu–Cu in the catalyst after electroreduction, and the Cu$^{I}$/Cu$^0$ ratio was unchanged after 5 min, which is consistent with the results of XPS study (Supplementary Fig. 21 and Supplementary Table 5). The transmission electron microscopy (TEM) and high-resolution transmission electron microscopy (HR-TEM) images of the catalyst with an electroreduction time of 5 min are presented in Fig. 2e–g, demonstrating that high density of grain boundaries (GBs) exist between Cu and Cu$_2$O. The oxygen vacancies on the interface of Cu and Cu$_2$O create new defect states located in the band gap, and the electrons on the defect states are easily excited, leading to the enhanced conductivity[49–51]. All the results above show that the Complex-1 changed into Cu and Cu$_2$O composite denoted as Cu–Cu$_2$O-1 (the number corresponds to the number of the complex) on Cu substrate after electroreduction, and the composition and morphology were stable after 5 min of reduction. As expected, the Cu–Cu$_2$O-1 film and Cu substrate in the Cu–Cu$_2$O-1/Cu also compacted tightly without any noticeable crack (Fig. 2b). In this work, the Cu-complexes on the Cu substrate acted as the precursor and template for the formation of the Cu–Cu$_2$O-1 catalyst with 3D structure on the Cu substrate.

**The electrocatalytic performance of Cu–Cu$_2$O/Cu.** The linear sweep voltammograms (LSVs) over Cu–Cu$_2$O-1/Cu electrode in CO$_2$ or N$_2$ saturated 0.1 M KCl aqueous electrolyte were determined (Supplementary Fig. 24), which shows reduction of CO$_2$ clearly. The dependence of current density on concentration of KCl solution saturated by CO$_2$ over Cu–Cu$_2$O-1/Cu electrode is provided in Supplementary Fig. 25, demonstrating that 0.1 M KCl solution was the optimal electrolyte.

The electrocatalytic performance of the Cu–Cu$_2$O-1/Cu electrode was further investigated by electrolysis of CO$_2$ at different applied potentials in 0.1 M KCl aqueous solution using a typical H$^-$ type cell, in which water acted as a proton source[36,52],

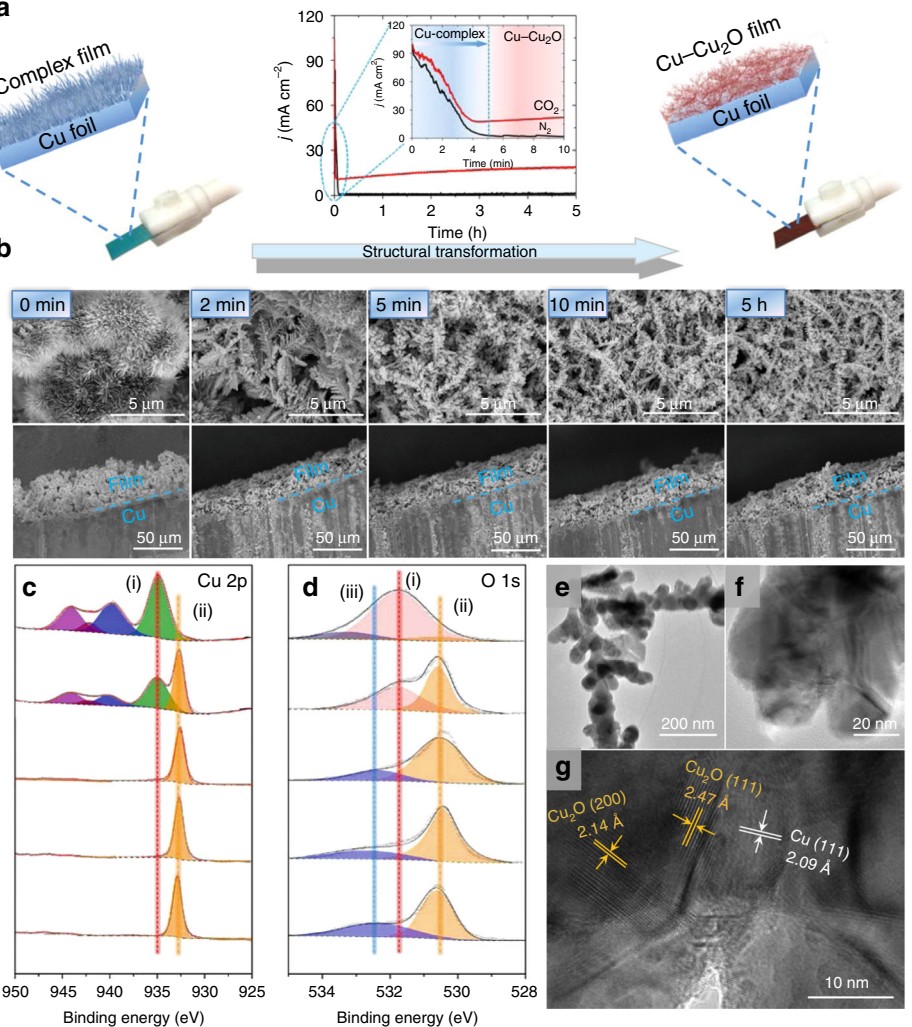

**Fig. 2** Decomposition of copper–cuprous oxide electrodes. **a** Reduction current density in 0.1 M KCl solution saturated with $CO_2$ and $N_2$ at applied potential of −0.4 V vs RHE, and the inserts show the current density at initial stage; **b** SEM images of the electrode with different electroreduction times (0 min, 2 min, 5 min, 10 min, 5 h); **c, d** Quasi in situ XPS spectra of the catalyst with different electroreduction times (0 min, 2 min, 5 min, 10 min, 5 h). Cu 2p photoelectron peak distribution: (i) $Cu^{II}$ of Complex-1 structure; (ii) $Cu^I/Cu^0$ of Cu–Cu$_2$O-1 structure; O 1s photoelectron peak distribution: (i) Oxygen of $Cu_xO$ on Complex-1 structure; (ii) Oxygen of $Cu_xO$ on Cu–Cu$_2$O-1 structure; and (iii) oxygen defect species; **e** TEM and **f, g** HR-TEM images of the Cu–Cu$_2$O-1 with electrorduction time of 5 h

and the results are given in Fig. 3. The experiment at each condition was repeated three times and the average values are used. In this work, the applied potential is referencing to the reversible hydrogen electrode (RHE) and the current density is calculated by geometric surface area. Under the reaction conditions, acetic acid and ethanol were the only liquid products detected by NMR spectroscopy, and $H_2$, CO and $CH_4$ were the gaseous product determined using gas chromatography (GC) (Supplementary Fig. 26 and Supplementary Table 7). The electrocatalytic performances of neat Cu foil, Cu foam obtained by direct electrodeposition without using ligand, $Cu_2O$ and CuO electrodes prepared using reported method[11,40,41,53] were also studied for comparison (Supplementary Fig. 27), and the results are also given in Fig. 3. Clearly, Cu–Cu$_2$O−1/Cu electrode had much better performance for $C_2$ product (acetic acid and ethanol) generation than other Cu-based electrodes. The applied potential and the overpotential over the Cu–Cu$_2$O-1/Cu electrode were much lower (Fig. 3a), and the FE was much higher (Fig. 3b). At applied potential of −0.4 V vs RHE, the overpotential on Cu–Cu$_2$O-1/Cu electrode was as low as 0.53 V (for acetic acid) and 0.48 V (for ethanol) (Fig. 3c, d). The current density and FE

for the $C_2$ product were 11.5 mA cm$^{-2}$ and 80.7% (48% for acetic acid and 32% for ethanol), respectively. Supplementary Table 8 presents the recent advances in reduction of $CO_2$ to $C_{2+}$ products. Previous report indicates that dendritic Cu catalysts are capable of reducing $CO_2$ to $C_2$ products. These desired structures can be synthesized through different methods[7,38,54]. However, the characteristics of the catalyst such as structure and oxidation state are difficult to be controlled, which lead to high overpotential of the $C_2$ products. Using molecular complex as an additive is another method to promote reduction of $CO_2$ to $C_2$ product[30–32]. However, as a precursor, the complex was generally prepared ex situ and used as a homogeneous additive in decomposition of the new structure. Compared with these methods, the Cu–Cu$_2$O catalyst synthesized in this work through in situ deposition/decomposition method exhibited controllable 3D structure, high charge transfer rate and lower overpotential toward liquid $C_2$ products. By comparing with the data reported (Supplementary Table 8), it can be known that this in situ electrosynthesized Cu–Cu$_2$O-1/Cu electrode had significantly lower applied potential and overpotential, higher current density, and higher FE for liquid $C_2$ products, especially in aqueous

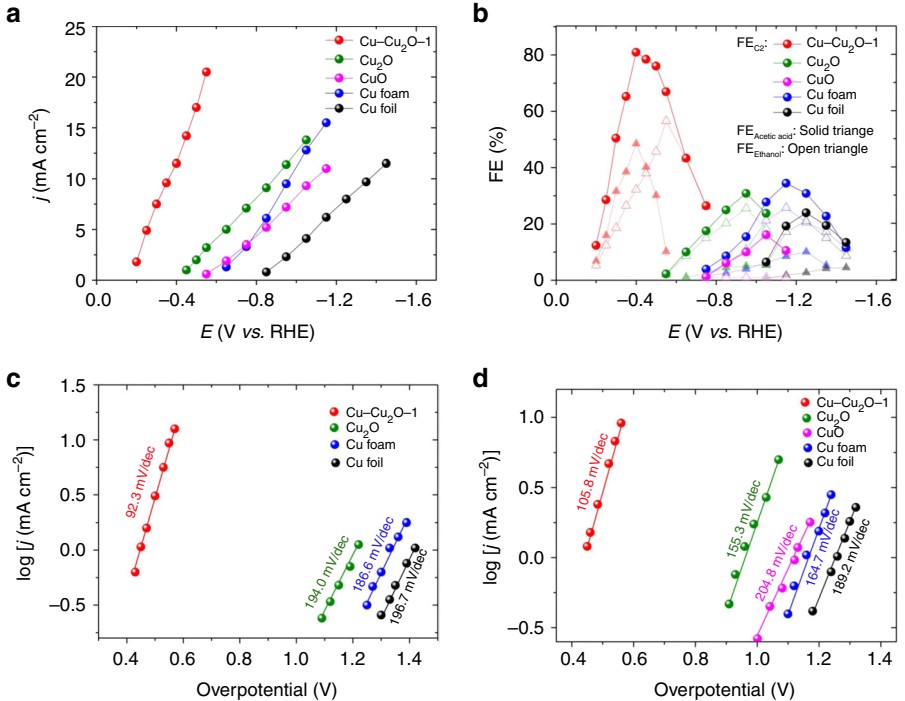

**Fig. 3** Carbon dioxide reduction over various copper-based electrocatalysts. **a** The total current density at different applied potentials; **b** Faradaic efficiencies of the $C_2$ products at different applied potentials; **c, d** Dependence of partial current density of acetic acid and ethanol on overpotential. Data were collected at room temperature and ambient pressure; electrolyte, 0.1 M KCl; $CO_2$ stream, 5 sccm

electrolyte. In addition, the Cu–Cu$_2$O-1/Cu electrode exhibited long-term stability in the electrolysis, which was known from the fact that the current density and FE did not change considerably with electrolysis time in 24 h, as shown in Supplementary Fig. 28. The result in Supplementary Table 9 indicates that the amounts of the $C_2$ products changes with applied potential and the amounts of the generated $C_2$ products increased almost linearly with increasing of electrolysis time (Supplementary Fig. 29). The turnover frequency (TOF) for $CO_2$ reduction reaction over the catalysts at different applied potentials are provided in Supplementary Table 10. The Cu–Cu$_2$O-1/Cu displayed best performance and the TOFs of acetic acid and ethanol were 17.0 and 15.6 $h^{-1}$, respectively.

Electrolyte also plays an important role in the $CO_2$ reduction. Recently, researchers have reported the reduction of $CO_2$ to $C_{2+}$ products in KCl aqueous electrolyte[6,55–57], and it was found to decrease the overpotential and increase the $C_2$ selectivity. The rise in local pH facilitated a higher amount of adsorbed *CO, which promotes their C–C coupling to $C_2$ products[56,58–62]. To investigate whether the local pH affected the product selectivity, we have also carried out experiments using KHCO$_3$ electrolyte (Supplementary Figs. 30–32 and Supplementary Table 11). The linear sweep voltammograms using $CO_2$ saturated KHCO$_3$ aqueous electrolytes of different concentrations (Supplementary Fig. 31a) illustrate that maximum current density was obtained in 0.5 M KHCO$_3$ solution. The electrolysis experiments showed that the current densities and selectivity to $C_2$ products in KCl electrolyte were higher than those in KHCO$_3$ electrolyte at all applied potentials (Supplementary Fig. 31c), indicating that the electrolytes influence the electrochemical reaction significantly. One of the main reasons may be that in concentrated KHCO$_3$ solution, sufficient HCO$_3^-$ is present to neutralize OH$^-$. Therefore, the formation of H$_2$ is favored and $C_2$ product is less preferred[58]. The result suggests that local pH influenced the

selectivity of the reaction, which is consistent with the results in the recent studies[6,54–57].

To confirm that $CO_2$ was carbon source of the product, we also conducted the blank experiments using N$_2$ to replace $CO_2$ in the electrolysis. The experiments showed that no product was formed in the electrolysis when using N$_2$. To further verify that the product was derived from $CO_2$ reduction, isotope⁻labeled $^{13}CO_2$ was used to study the reaction over Cu–Cu$_2$O-1/Cu. From $^1$H NMR spectra in Supplementary Figs. 33 and 34, we can only see the H signal of $^{13}CH_3$ group on the acetic acid and ethanol, which splits into two peaks by the coupling with $^{13}$C atom. Moreover, the $^{13}$C NMR spectra showed strong $^{13}$C signals as $^{13}CO_2$ was used as the feedstock (Supplementary Fig. 35). The results further confirm that all the carbon atoms in the product were from $CO_2$, which is consistent with the conclusion obtained from the blank experiment using N$_2$.

**Investigation into the catalytic ability of Cu–Cu$_2$O/Cu.** There are several reasons for the outstanding performance of the Cu–Cu$_2$O-1/Cu electrode in the electrocatalytic reduction of $CO_2$ to the $C_2$ products. First, the Cu–Cu$_2$O-1 catalyst and Cu substrate contacted very well in the electrode. This structure can reduce or eliminated the contacting resistance between the catalyst and the Cu substrate, which is favorable to reducing the overpotential and applied potential. In order to get some evidence to support this point, some control experiments were designed. We prepared Complex-1 by solvothermal method using the same ligand and fabricated the complex electrode by loading the complex suspension onto the carbon paper (CP) using conventional drop-casting method (Supplementary Figs. 36 and 37), and the complex was also reduced to Cu–Cu$_2$O composite by electroreduction to obtain Cu–Cu$_2$O-1/CP electrode as known from the XPS characterization (Supplementary Fig. 38). Supplementary

Fig. 39 compared the CV curves obtained using the Cu–Cu$_2$O-1/Cu and Cu–Cu$_2$O-1/CP electrodes prepared by the two methods. Supplementary Fig. 40 and Supplementary Table 12 show the electrolysis results over the two electrodes at the optimized potential. The applied potential and overpotential over the Cu–Cu$_2$O-1/Cu electrode was much lower than that over Cu–Cu$_2$O-1/CP electrode. Electrochemical impedance spectroscopy (EIS) study showed that the film resistance ($R_f$) between the catalyst and substrate in the Cu–Cu$_2$O-1/Cu electrode was much smaller than that in the Cu–Cu$_2$O-1/CP. Moreover, the charge transfer resistance ($R_{ct}$) between the electrolyte and electrode surface of Cu–Cu$_2$O-1/Cu electrode was also much smaller than that of Cu–Cu$_2$O-1/CP electrode system (Supplementary Fig. 41 and Supplementary Table 13). The smaller $R_f$ and $R_{ct}$ result in the lower applied potential and overpotential. Second, the Cu–Cu$_2$O-1 catalyst in Cu–Cu$_2$O-1/Cu had 3D dendritic structure. The 3D structure results in abundant exposed active sites (i.e., grain boundaries, oxygen vacancies), leading to high activity of the Cu–Cu$_2$O-1 electrocatalyst. To verify the importance of 3D structure, we performed the CO$_2$ reduction using the Cu–Cu$_2$O-1/Cu electrodes obtained from in situ reduction of the Complex-1/Cu prepared at different electrodeposition times (Fig. 1b and Supplementary Fig. 42). Supplementary Fig. 42 shows that the current density increased continuously with electrodeposition time. This can be explained by the fact that the amount of Cu–Cu$_2$O-1 catalyst in the Cu–Cu$_2$O-1/Cu electrodes increased with electrodeposition time. The FE increased with electrodeposition time at beginning, and then became nearly independent of electrodeposition time. Figures 1b and 2b show that the 3D structure gradually formed with increasing deposition time. A reasonable interpretation of the result is that 3D structure provides more opportunity for the C–C coupling because of the longer residue time of the reaction species. When the film of 3D structure was thick enough, the residue time was long enough for the intermediates to form the C$_2$ product, and thus the FE was not changed obviously with the thickness of the Cu–Cu$_2$O-1 film. Third, the ligand in the complex is crucial for the formation of precursor, which is important for the construction of Cu–Cu$_2$O structure (Supplementary Fig. 19). This can be known from the fact that the FE over other Cu-based electrodes, such as metallic Cu, electrodeposited Cu foam, Cu$_2$O, and CuO was much lower (Fig. 3, Supplementary Fig. 27). In addition, the density functional theory (DFT) calculation and experimental study also showed that Cu–Cu$_2$O is preferred electrocatalyst for production of C$_2$ products[57,63]. The synergy between the active surfaces of Cu$^I$/Cu$^0$ improves significantly both CO$_2$ activation and CO dimerization to generate C$_2$ products. The results also showed that use of the Cu-complex as the precursor is crucial for the outstanding performance because the efficiency of Cu catalyst obtained by direct electrodeposition was much poorer.

**The electrocatalytic performance of other Cu–Cu$_2$O/Cu electrodes**. In order to verify the versatility of the method, we also prepared Cu–Cu$_2$O/Cu electrodes using other ligands, including 1,2,4-H$_3$BTC (2), 1,3,5-H$_3$BTC (3), 1,2,3-H$_3$BTC (4), 2,6-H$_2$PyDC (5), and 1,4-H$_2$BDC (6). The complexes on Cu substrate are denoted as Complex-2, Complex-3, Complex-4, Complex-5, Complex-6, respectively. The complexes were also characterized using different methods, and the results are given in Supplementary Figs. 1–11, Supplementary Figs. 13–18, and Supplementary Tables 1–4. The corresponding electrodes are represented by Cu–Cu$_2$O-2/Cu, Cu–Cu$_2$O-3/Cu, Cu–Cu$_2$O-4/Cu, Cu–Cu$_2$O-5/Cu, Cu–Cu$_2$O-6/Cu, and the characterization results are provided in Supplementary Figs. 19–21 and Supplementary Table 5. All the results showed that 3D dendritic Cu$_2$O–Cu films

on Cu electrodes could also be prepared using these ligands, and the detailed discussion on this are provided in the supporting information. Subsequently, the as-prepared Cu–Cu$_2$O/Cu electrodes were used for CO$_2$ reduction in 0.1 M KCl aqueous electrolyte, and the results are given in Fig. 4, Supplementary Figs. 43–45. Generally, these Cu–Cu$_2$O/Cu electrodes could also promote CO$_2$ electroreduction to C$_2$ products at very low applied potential and overpotential with high selectivity in the aqueous electrolyte. At −0.4 V vs RHE, the FEs of C$_2$ products over Cu–Cu$_2$O-2/Cu, Cu–Cu$_2$O-3/Cu, Cu–Cu$_2$O-4/Cu, Cu–Cu$_2$O-5/Cu, and Cu–Cu$_2$O-6/Cu were 67.6%, 56.7%, 54.4%, 50.2%, and 41.8%, respectively. The results suggest that the strategy to prepare 3D dendritic Cu–Cu$_2$O/Cu electrodes for highly efficient electroreduction of CO$_2$ to C$_2$ product is versatile. Among all the electrodes, Cu–Cu$_2$O-1/Cu has the best activity, which may results from the lowest charge transfer resistance between electrolyte and electrode (Supplementary Table 13B). In addition, the largest surface roughness factor of the Complex-1 precursor (Supplementary Fig. 11 and Supplementary Table 4) favored the generation of Cu–Cu$_2$O with abundant exposed active sites. All these factors can enhance the catalytic activity of the electrode.

**Mechanism**. With the aim to understand the mechanistic pathway toward the formation of acetic acid and ethanol by CO$_2$ reduction, some control experiments were conducted in the presence of the possible reaction intermediates, such as CO, formic acid, formaldehyde, acetaldehyde, and acetic acid. Supplementary Table 14 presents the production rates of acetic acid and ethanol obtained in these experiments. The results indicate that CO, formaldehyde and acetaldehyde are three important intermediates in the C$_2$ product pathway. Supplementary Fig. 46A shows the IR spectra of the electrolyte after different electrolysis times. The intensity of acetic acid and ethanol increased with increasing electrolysis time, indicating that the amount of C$_2$ products generated increased with electrolysis time, suggesting the formation of acetic acid and ethanol from electroreduction of CO$_2$. We also conducted the IR study in the presence of possible reaction intermediates (Supplementary Fig. 46B). The result indicates that CO, formaldehyde, and acetaldehyde are the possible intermediate in the mechanistic pathway. This conclusion is consistent with the result in Supplementary Table 14. Supplementary Scheme 2 gives the possible mechanistic pathway for the electrocatalytic production of acetic acid and ethanol. The three key steps are (i) CO$_2$ activation; (ii) C$_1$ product formation, which was found to compete with C$_2$ products; (iii) CO dimerization, which can be important *COCHO or *COCO intermediate for C$_2$ products formation. Subsequently, the reaction is divided into two paths. For formation of ethanol, both *COCHO and *COCO can be considered as the precursors. The C–C bond is subsequently formed and ethanol is generated via reduction of CH$_3$CHO intermediate. For acetic acid, the intermediate *COCHO is further reduced at the electrode surface to form −CH$_3$COO$^-$ species. This tentative mechanistic pathway is able to explain the experimental observations discussed above.

**Discussion**

In summary, 3D dendritic Cu–Cu$_2$O/Cu electrodes for CO$_2$ reduction can be prepared by electroreduction of in situ deposited Cu-complex film on Cu substrate successfully. They had outstanding performance for electrochemical reduction of CO$_2$ to C$_2$ product in KCl aqueous solution. The results show that reducing contacting resistance between the catalysts and the substrate in the electrodes, increasing exposed active sites in the 3D structure and chemical composition of Cu–Cu$_2$O composition are favorable to enhance the efficiency for the reduction of CO$_2$ to C$_2$

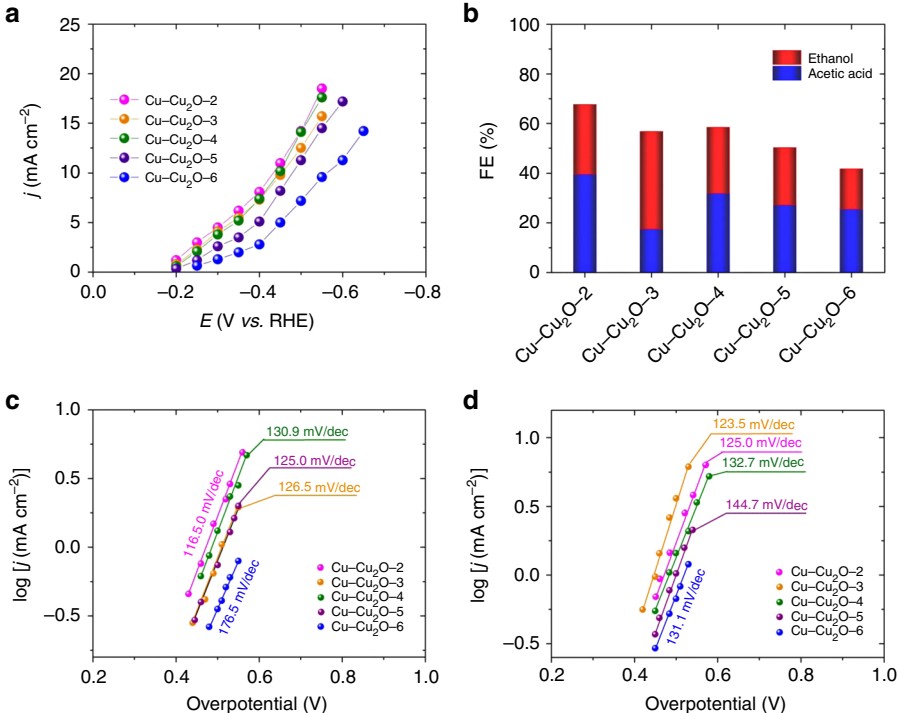

**Fig. 4** Carbon dioxide reduction over copper–cuprous oxide electrodes prepared using ligands two to six. **a** The total current density at different applied potentials; **b** Faradaic efficiencies of the C₂ products at applied potential of −0.4 V vs RHE. **c**, **d** Dependence of partial current density of acetic acid and ethanol on overpotential, respectively. Data were collected at room temperature and ambient pressure; electrolyte, 0.1 M KCl; CO₂ stream, 5 sccm

products. This work opens a simple way to efficient transformation of $CO_2$ into valuable chemicals or liquid fuels, and the findings are helpful for designing other electrodes for efficient electroreduction of $CO_2$. We believe that this work will trigger many interesting work in the future.

## Methods

**Materials**. $CO_2$ (Beijing Beiwen Gas Chemical Industry Co., Ltd., research grade) had a purity of 99.999% and used as received. $^{13}CO_2$ (99 atom% $^{13}C$, <1 atom%) was purchased from Beijing Gaisi Chemical Gases Center. Pyromellic acid (1,2,4,5-H₄BTC, 96%), 1,2,4-benzenetricarboxylic acid (1,2,4-H₃BTC, 98%), 1,3,5-benzenetricarboxylic acid (1,3,5-H₃BTC, purity > 99%), 1,2,3-benzenetricarboxylic acid (1,2,3-H₃BTC, 95%), 2,6-pyridinedicarboxylic acid (2,6-H₂PyDC, 99%), tetramethylammonium bromide (TMAB, 99%), and tetrabutyl ammonium perchlorate (TBAP, 99.9%) were obtained from J&K Scientific Ltd. 1,4-Bezenedicarboxylic acid (1,4-H₂BDC, 99%) was purchased from TCI. Cu(NO₃)₂·3H₂O (A. R. grade), acetonitrile (AcN, A. R. grade), methanol (A. R. grade), dimethylformamide (DMF, A. R. grade), acetone (A. R. grade), and Cu foil (0.2 mm in thickness, purity > 99.99%) were provided by Sinopharm Chemical Reagent Co., Ltd, China. Potassium chloride (KCl, 99.997%) was purchased from Alfa Aesar China Co., Ltd, which was purified by recrystallization three times before use. Toray carbon paper (CP, TGP-H-60, 19 × 19 cm) and Nafion N-117 membrane (0.180 mm thick, ≥ 0.90 meg/g exchange capacity) were purchased from Alfa Aesar China Co., Ltd. Polytetrafluoroethylene (PTFE, 60 wt% aqueous solution) was purchased from Sigma-Aldrich Co. LLC. Tetrabutylammonium hexfluorophosphate (TBAPF₆, >98%) was provided by the Centre of Green Chemistry and Catalysis, Lanzhou Institute of Chemical Physics, Chinese Academy of Sciences.

**In situ electrosynthesis of the Complex-1 to 5 on Cu substrate**. The structures of the ligands used are shown in Supplementary Scheme 1. The procedures were similar to that used for the electrosynthesis of metal organic frameworks[40,41]. Typically, two Cu foils with dimension of 0.2 mm × 10 mm × 10 mm were used for the working and counter electrode with a gap of 1 cm and the electrochemical experiments could be controlled by a DC mode on a galvanostat/potentiostat (CS310, Wuhan Corrtest Instrument Co., China). The electrolyte consisted of 25 mL of ethanol/water (75:25 vol%) solution, 5–15 mg MTAB (99.9%, supporting electrolyte), and 5 mg Cu-complex-1 (or 15 mg Complex-2, or 10 mg Complex-3, or 5 mg Complex-4, or 15 mg Complex-5, or 15 mg Complex-6). The electrosynthesis was performed by applying a potential difference of 9.0 V between the Cu foil electrodes in 70 °C electrolyte. Cu-Complex was in situ formed at the anode.

**In situ electrosynthesis of Complex-6 on Cu substrate**. The procedures for the synthesis of Complex-6 film with organic linker 1,4-BDC was similar to that discussed above. The electrolyte was DMF solvent with 20 g/L of linkers and 15 mg of MTAB was used as the supporting electrolyte. Before electrodeposition, the electrolyte was stirred at 70 °C for several minutes before a clear solution was obtained. The anodic deposition was performed by applying a potential difference of 9.0 V between the two Cu foils.

**Synthesis of Complex-1 by solvothermal method**. The procedures to synthesize Complex-1 by thermal method were similar to that for preparing HKUST-1[45]. A ethanol/water (75:25 vol%) solution was used as the solvent. In a typical experiment, 1,2,4,5-H₄BTC (0.12 g, 0.5 mM) was dissolved in 20 mL of ethanol, and Cu(NO₃)₂·3H₂O (0.242 g, 1.0 mM) was dissolved in 5 mL of H₂O. The Cu(NO₃)₂·3H₂O solution was slowly added to the 1,2,4,5-H₄BTC solution with stirring at room temperature. Then 2 mL of DMF was added to the mixture solution with stirring. The mixture was transferred into a Teflon-lined autoclave and the reaction occurred at 90 °C under hydrothermal condition for 48 h. After cooling to room temperature, the solid was collected and washed with H₂O and ethanol. The obtained Cu-Complex-1 was dried in a vacuum oven at 80 °C for 24 h.

**Synthesis of Complex-3 by solvothermal method**. The procedures to synthesize the Complex-3 with organic linker 1,3,5-H3BTC were similar to that reported previously[45]. In a typical experiment, Cu(NO₃)₂·3H₂O (2.6 g, 10.7 mM) was dissolved in 30 mL of H₂O in a flask. 1,3,5-H3BTC (0.68 g, 3.2 mM) was dissolved in 30 mL of ethanol. The Cu(NO₃)₂·3H₂O solution was slowly added to the 1,3,5-H3BTC solution with stirring at room temperature. The solution became turbid with formation of precipitate. DMF (2 mL) was added to the mixture, and then the combination was transferred to a Teflon-lined autoclave and allowed to be reacted at 80 °C for 24 h. After cooling to room temperature, the solid was collected and washed with H₂O and ethanol. The obtained Cu-Complex was dried in a vacuum oven at 80 °C for 24 h.

**Fabrication of the electrodes using Cu-Complexes synthesized by solvothermal method**. The complex electrodes were prepared by loading Cu-Complex suspension onto the carbon paper (CP) using drop-casting method. Briefly, the as-prepared Cu-Complex was dispersed in acetone, then suitable amount of 6.0 wt% PTFE aqueous solution was added to the as prepared Cu-Complex dispersion. The final mixture contained 10 mg/mL complex and 0.6 wt% PTFE, which was ultrasonicated for 30 min to form a uniform suspension. Then 0.1 mL of the suspension

was loaded on the 1.0 cm × 1.0 cm CP. The electrodes were dried in an oven at 60 °C for 1 h before the electrochemical experiment.

**Electrodeposition of $Cu_2O$ on Cu foil.** Electrochemical growth of $Cu_2O$ film was accomplished by using a conventional three-electrode cell using the method reported[11]. The copper foil substrate was used as working electrode, a platinum sheet as counter electrode, and Ag/AgCl as reference electrode. The electrodeposition was carried out cathodically using the solution containing 0.02 M $CuSO_4$ and 0.34 M DL-Lactic acid (85%) at pH = 9 adjusted using 2 M NaOH. Deposition were carried out at 70 °C and −0.95 V vs Ag/AgCl for 30 min.

**Electrodeposition of CuO on Cu foil.** CuO thin films were grown by electrodeposition technique using potentiostatic method reported[53]. A standard three-electrode cell was used for the electrodeposition. The copper foil substrate was used as working electrode, a platinum sheet as counter electrode, and a saturated calomel electrode (SCE) as the reference electrode. The electrodeposition of CuO was carried out cathodically from an aqueous bath composed of $CuSO_4$ (0.03 M) and L(+) tartaric acid (0.03 M). The deposition of CuO thin film was carried out at a deposition potential of −650 mV vs SCE. The solution pH was adjusted to 10–12 by the addition of 2 M NaOH solution. The deposition was carried out at 75 °C for 30 min.

**Direct electrodeposition of Cu foam on Cu foil.** The method was similar to that for synthesis of Cu-Complex[40,41], and the main difference was that the ligand was not used. A Cu foam on Cu foil was electrodeposited in a solution containing 0.25 M $CuSO_4$, 25 mL of ethanol/water (75:25 vol%) solution, 5–15 mg MTAB (99.9%, supporting electrolyte) at ambient temperature (25 °C). Prior to experiment, the electrolyte was deoxygenated by high-purity nitrogen for at least 30 min. Cu foam was deposited onto Cu foil by applying a constant potential of 9.0 V between two Cu foil electrodes in the electrolyte mentioned above.

**Material characterization.** Powder X-ray diffraction (XRD) patterns were acquired with a X-ray diffractometer (Model D/MAX2500, Rigaka) with Cu-Kα radiation, and the scan speed was 5°/min. X-ray photoelectron spectroscopy (XPS) analysis was conducted on the Thermo Scientific ESCALab 250Xi (USA) using 200 W monochromatic Al Kα radiation. The 500 μm X-ray spot was used for XPS analysis. The base pressure in the analysis chamber was about $3 \times 10^{-10}$ mbar. The contents of Cu and C in the complex were determined by induced coupled plasma optical emission spectrometer (ICP-OES, VARIN VISTA-MPX). The morphologies of materials were characterized by a HITACHI S-4800 scanning electron microscope (SEM) and a JEOL JEM-2100F high-resolution transmission electron microscopy (HR-TEM).The porosity properties and surface areas of the materials were obtained from nitrogen adsorption–desorption isotherms determined using a Micromeritics ASAP 2020 M system. For XRD, ICP-OES, gas adsorption–desorption measurements, in order to get enough sample, at least 10 electrodes were prepared at the same conditions and the catalyst films were scraped and collected for characterization.

The FT-IR spectra were collected at a resolution of 4 cm$^{-1}$ on a Bruker Vector 27 spectrophotomfeter in the 400–4000 cm$^{-1}$ region. The IR spectra of samples were measured by the conventional KBr pellet method. To get enough sample, at least 10 electrodes were prepared at the same conditions and the complex films were scraped and collected.

The Quasi in situ X-ray photoelectron spectra (XPS) were measured on an AXIS ULTRA DLD spectrometer with AlKα resource (hv = 1486.6 eV). The protective sample transfer procedure was similar to that reported in the literature[64]. For investigating the evolution of Cu species in the reaction process, catalysts were electrolyzed with different times in the $CO_2$-saturated electrolytes. After that, the samples were immediately immersed in Ar-prepurged acetone and then transferred into an Ar-filled glove box to protect the catalyst from directly contacting of air. Then, the samples was cleaned, dried and cut into 3 × 3 mm and glued on a stage with a double sided adhesive. The stage was evacuated into vacuum to prevent the sample to be oxidized in the air. The subsequent testing processes were the same as that of the common X-ray photoelectron spectroscopy. The contaminated carbon C 1s signal at 284.8 eV was used to calibrate binding energies. The spectra were deconvolved with XPS PEAK 4.1 software by subtracting the Shirley background and applying the Lorentzian–Gaussian function. The modified Auger parameter ($\alpha_{Cu}$) was calculated according to the following equation:

$$\alpha_{Cu} = E_B + E_K \qquad (1)$$

where $E_B$ is the binding energy of the Cu $2p_{3/2}$ core level and $E_K$ is the kinetic energy of the Cu LMM Auger electron.

X-ray absorption fine structure spectroscopy (XAFS) measurements were taken at 1W2B beamline of Beijing Synchrotron Radiation Facility (BSRF), China. The Cu K-edge (8979 eV) XAFS spectra were collected under electrochemical environment using the fluorescence detection method. The EXAFS data were processed according to the standard procedures using the ATHENA module implemented in the IFEFFIT software packages. The Cu K-edge X-ray absorption spectra of the catalysts were collected after reaction at different times. The Cu K-

edge X-ray absorption spectra of Cu-complexes, CuO, $Cu_2O$, and Cu foil were collected for reference.

Small angle X-ray scattering (SAXS) experiments were carried out at Beamline 1W2A at the Beijing Synchrotron Radiation Facility. The apparatus and the procedures were similar to that used in previous work[52,65]. The data were collected using a CCD detector (MAR) with maximum resolution of 3450 × 3450 pixels. The wavelength of the X-ray was 1.54 Å, and the distance of the sample to detector was 1.31 m. In a typical experiment, the sample was added into the sample cell, and the X-ray scattering data were recorded. The 2-D SAXS images were obtained from the detector and then transformed into the profiles of intensity (I) vs wavevector (q) by the software FiT2D[52,65]. To get enough sample, at least 10 electrodes were prepared at the same conditions and the complex films were scraped and collected.

For XRD, ICP-OES, $N_2$ adsorption–desorption, FT-IR, XAFS, and SAXS measurements, in order to get enough sample, at least 10 electrodes were prepared at the same conditions and the catalyst films were scraped and collected for characterization.

**Crystal data and structure refinement for Complex-1, Complex-4, and Complex-5.** In total, 0.05 mmol of 1,2,4,5-$H_4$BTC, or 1,2,3-$H_3$BTC or 2,6-$H_2$PyDC was dissolved in 20 mL ethanol, and then a solution of 0.05 mmol $Cu(NO_3)_2$ in 5 mL $H_2O$ was added to the mixture. As the solvent evaporating, blue crystals were obtained after solvent evaporation. Then, it was washed with acetone and air dried. The product was recrystallized from ethanol–$H_2O$ (4:1).

Crystallographic measurements for Complex-1, Complex-4, and Complex-5 were made using a Bruker APEX area-detector diffractometer. The intensity data were collected using graphite monochromataed Mo κα radiation. The structures were solved by direct methods and refined by full-matrix least-squares techniques on F2. Structure solution and refinement was accomplished using SIR97, SHELXL97, and WINGX. The all H atoms were positioned geometrically and allowed to ride on their parent atoms. The molecular structure plots were prepared by using ORTEPIII.

Crystal data and structure refinement are given in Supplementary Figs. 13–18. Anisotropy thermal parameters, structure factors, full lists of bond distances, bond angles, and torsion angles are given in Supplementary Table 15. Selected bond lengths and angles are given in Supplementary Tables 16–21. The CCDC number of Complex-1, Complex-4, and Complex-5 were 1936102, 1936103, and 193104, respectively.

**pH value measurements.** The pH values of gas-free, $N_2$-satuarated, and $CO_2$-saturated electrolyte solutions (0.1 M KCl) were determined by an electronic pH meter (METTLER TOLEDO FE-20K). To determine the pH of the $CO_2$-saturated electrolyte solution at ambient temperature, $CO_2$ was bubbled into 20 mL solution in a beaker (25 mL/min) under stirring. The pH value of the solution was monitored at different times, and the value was recorded when the value became unchanged with time.

**Linear sweep voltammetry and pH value measurements.** An electrochemical workstation (CHI 6081E, Shanghai CH Instruments Co., China) was used for all $CO_2$ reduction experiments. The apparatus and procedures were similar to that used previously[52]. The linear sweep voltammetry (LSV) measurements were carried out in a single compartment cell with three-electrode configuration, which consisted of a working electrode, a platinum gauze auxiliary electrode and Ag/AgCl reference electrode. Aqueous solution of KCl or KHCO$_3$ was used as cathode electrolyte. Before each set experiment, the electrolyte was bubbled with $N_2$ or $CO_2$ for at least 30 min to form $N_2$ or $CO_2$ saturated solution. The LSV measurement in gas-saturated electrolyte was conducted at a sweep rate of 20 mV/s in the potential range of −0.2 to −1.1 V vs Ag/AgCl. For a catalyst prepared using the Cu-complex as the precursor, the complex was first electroreduced for 5 min at 0.4 V vs RHE to form Cu–$Cu_2O$ catalyst.

In this work, all potentials were referencing to RHE by considering the pH of the solution by Eq. (2)[66].

$$E \text{ vs RHE} = E \text{ vs Ag/AgCl} + 0.197\,\text{V} + 0.0592 \times \text{pH (V)} \qquad (2)$$

**Thermodynamic potentials.** The overpotential is the difference between the applied potential and the thermodynamic potential for the reaction of interest. Thermodynamic potentials for the reactions of $CO_2$ to acetic acid and $CO_2$ to ethanol were calculated, and the values are 0.125 and 0.084 V, respectively. The data are calculated from the standard molar Gibbs energy of formation at 298.15 K[67,68] and the method is discussed below. The calculation assumes that gases are at 1 atm and liquids are in their pure form.

$$2\,CO_2 + 6\,H_2O + 8e^- \rightarrow CH_3COOH_{(l)} + 8\,OH^- \qquad 596.4 \text{ kJ mol}^{-1} \qquad (3)$$

$$CH_3COOH_{(l)} \rightarrow CH_3COOH\,(1\,\text{M}) \qquad -29.4 \text{ kJ mol}^{-1} \qquad (4)$$

Therefore, compared with hydrogen evolution reaction (HER) in aqueous electrolyte solutions (Eq. (5)), the primary reactions that occurred on the electrode

in aqueous solution at pH 7.0 vs standard hydrogen electrode (SHE) (Eq. (6)) is:

$$2H_2O + 2e^- \rightarrow 2OH^- + H_2 \quad E = -0.41 \text{ V vs SHE at pH 7.0} \quad (5)$$

$$2CO_2 + 6H_2O + 8e^- \rightarrow CH_3COOH\,(1\text{ M}) + 8OH^- \quad E = -0.289 \text{ V vs SHE at pH 7.0} \quad (6)$$

Applying the Nernst equation (Eq. (7)), the equilibrium potential is estimated to be:

$$E^0 = -0.289 \text{ V} + \frac{RT\ln(10)}{F}\text{pH} = 0.125 \text{ V vs RHE at pH 7.0} \quad (7)$$

Therefore, using the same method, $E^0$ for the half reaction of $CO_2$ to acetic acid and $CO_2$ to ethanol are:

$$2CO_2 + 8H^+ + 8e^- \rightarrow CH_3COOH + 2H_2O \quad E^0_{\text{Acetic acid}} = 0.125 \text{ V vs RHE} \quad (8)$$

$$2CO_2 + 12H^+ + 12e^- \rightarrow CH_3CH_2OH + 2H_2O \quad E^0_{\text{Ethanol}} = 0.084 \text{ V vs RHE} \quad (9)$$

**Electrochemical impedance spectroscopy measurements**. The experimental apparatus was the same as that for LSV measurements. The experiment was carried out in $CO_2$ saturated 0.1 M KCl solution. The impedance spectra was recorded at open circuit potential (OCP) with an amplitude of 5 mV of $10^{-2}$–$10^{-5}$ Hz. The data obtained from the electrochemical impedance spectroscopy (EIS) measurements were fitted using the Zview software (Version 3.1, Scribner Associates, USA).

**Electrochemical capacitance measurements**. Cyclic voltammetry (CV) was used for the electrochemical capacitance measurements. The experiments were performed by collecting cyclic voltammograms in 0.1 M TBAPF$_6$ in MeCN solution at scan rates from 50 to 400 mV/s. The experiment was performed in three-electrode configurations, in which a silver wire was used as quasi-reference electrode and Pt gauze was used as counter electrode. Data obtained over a potential range of 40 mV around the open circuit potential of Cu–Cu$_2$O-1/Cu electrode is shown in Fig. S13. Plot of the current density from the quasi-square CV curves, calculated at the open circuit potential, as a function of scan rate. The surface roughness factor was calculated from the capacitance data[4,42].

**CO$_2$ reduction electrolysis and product analysis**. Controlled potential electrolysis (CPE) was carried out. The electrolysis experiments were conducted at 25 °C in a typical H-type cell, which was similar to that used in our previous works for $CO_2$ electrochemical reduction[52]. The as synthesized Cu-Complex electrodes were used as the working electrode. The Ag/AgCl (saturated KCl) was used as the reference electrode and the Pt gauze was used as counter electrode. The cathode and anode compartments were separated through a Nafion 117 proton exchange membrane. A KCl or KHCO$_3$ aqueous solution was used as cathode electrolyte. H$_2$SO$_4$ aqueous solution (0.5 M) was used as anodic electrolyte. Electrochemical reduction of $CO_2$ was carried out at ambient temperature. Under the continuous stirring, $CO_2$ was bubbled into the catholyte (25 mL/min) for 60 min before electrolysis. After that, potentiostatic electrochemical reduction of $CO_2$ was carried out with $CO_2$ bubbling (5 mL/min). The gaseous product was collected and analyzed by gas chromatography (GC, HP 4890D), which was equipped with FID and TCD detectors using helium as the internal standard. $^1$H NMR and $^{13}$C NMR measurements of products were performed on a Bruker Avance III 400 HD spectrometer. The samples were measured using DMSO-d$_6$ as a lock solution and TMS as an internal standard. To identify the reduction products, $^1$H NMR spectroscopy was carried out to quantify the liquid products after electrolysis of desired time at each given potential. The current density and Faradaic efficiencies (FEs) of the products were calculated using the amounts of the products obtained from GC and $^1$H NMR analysis.

Similarly, all potentials were referencing to RHE using the pH of the $CO_2$-saturated KCl solution.

After the quantification, the Faradaic efficiency (FE) toward each product were calculated as follows[58]:

$$FE(\%) = \frac{\text{amount of the product} \times n \times F}{C} \times 100 \quad (10)$$

where $n$ is number of moles of electrons to participate in the Faradaic reaction, $F$ is the Faraday constant (96485 C mol$^{-1}$), and $C$ is the amount of charge passed through the working electrode.

**Turnover frequency measurements**. TOF for $CO_2$ reduction reaction over the catalysts was calculated on the basis of amounts of products, electrolysis time, the moles of Cu species on the electrode determined gravimetrically. The moles of Cu species were assumed to the same as that in the complexes precursors. To get the reliable amount of the complex, at least 10 Cu-complex/Cu electrodes were prepared, and the mass of the complex was calculated from the mass difference before and after electrodeposition. The moles of Cu was known from the molecular weight of the complex (Supplementary Table 15) and mass of the complex in the complex/Cu electrode (Supplementary Table 3).

**Tafel analysis**. The Tafel plots were constructed from the partial current density and the overpotentail. The overpotential was the difference between the applied potential and the thermodynamic potential for the reaction of interest[4]. As mentioned above thermodynamic potentials for the reactions of $CO_2$ to acetic acid and $CO_2$ to ethanol are 0.125 and 0.084 V, respectively.

**Reporting summary**. Further information on research design is available in the Nature Research Reporting Summary linked to this article.

## Data availability

All data supporting the findings of this study are available from the corresponding author on request.

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

## Acknowledgements

The authors thank the National Key Research and Development Program of China (2017YFA0403101), National Natural Science Foundation of China (21773267, 21673248, 21733011) and the Chinese Academy of Sciences (QYZDY-SSW-SLH013). SAXS experiments were carried out at Beijing Synchrotron Radiation Facility.

## Author contributions

Q.G.Z., J.M. and B.X.H. proposed the project, designed the experiments and wrote the paper; Q.G.Z. performed the whole experiments. X.F.S. conducted the $^{13}C$ measurements. D.X.Y. performed FT-IR, XAFS and the XPS tests. L.R.Z. and J.Z. assisted in analyzing the XAFS results. X.C.K. and Z.H.W. assisted in analyzing the SAXS data. X.F.S., D.X.Y. and X.C.K. also performed the analysis of experimental data; B.X.H. supervised the whole project.

## Additional information

**Competing interests:** The authors declare no competing interests.

