## [Peer Review File · Nature Communications]

Reviewers' comments:

Reviewer #1 (Remarks to the Author):

This manuscript is of very high quality. The authors have done a substantial revision to the previously submitted version, which I highly appreciate. Although some of the earlier conclusions/claims have been modified/updated, the novelty and significance of this work still stands. It is particularly important that the authors have figured out the real catalytic sites via meticulous characterization. Controlled experiments were also carried out to interrogate the reaction mechanism. All my concerns for the previous version have been properly addressed. I recommend acceptance.

Two minor points:

While reading the experimental procedure, it was at least vague to me how the electrode was protected from being exposed to air during the transfer from the electrochemical cell to the XPS chamber. It is known that reduced Cu surface is sensitive to air (especially when the particle size is small) and that oxidized Cu species play important roles in the catalysis of CO₂ reduction. I thus suggest the authors elaborate on that. See for example: J. Phys. Chem. C 2018, 122, 2848-2853.

Judging from the potential-dependent product selectivity, acetic acid and ethanol seem to be competing for electrons. Is it possible that acetic acid is an intermediate for ethanol formation? Can the catalyst directly reduce acetic acid?

Reviewer #2 (Remarks to the Author):

The authors have dramatically revised their manuscript to answer reviewer concerns. Perhaps most transformative is their more comprehensive characterization of the electroactive material from the electrodeposited film which they now suggest is a Cu/CuO dendritic structure. They have attempted to characterize what they consider a "molecular" film via spectroscopic techniques, and they then electrochemically degrade this to a Cu/CuO by applying a sufficiently negative potential. The resulting active CO₂RR films are ligand and time dependent, suggesting that the ligand-metal structure of the initial electrodeposit helps template the final catalytically-active structure. The authors investigate this further for CO₂RR and show good activity for C₂ product formation.

I support publication of this manuscript in Nature Communications based on the characterization of their active material as a structured Cu catalyst and their empirical results involving C₂ product formation. However, I feel that some of the introduction is no longer relevant for their studies, and that the authors neglected to mention the effect of local pH gradients in their discussion of catalytic mechanism—this is a very important discussion in the CO₂ reduction literature on heterogeneous nanostructured materials! The authors should address the following comments prior to publication:

1. The authors should do a much broader literature search of the electrochemical decomposition of molecular precursors for structured electrocatalytic films to better put their system into the broader context of the most relevant literature. For instance, in lines 78-83 they state this is the first example of "conversion of CO₂ using metal organic complex electrocatalysts fabricated by electrosynthesis method." They then reference a MOF structure as a comparable structure used in sensors and electrical devices. A MOF is not the best description of their system based on their characterization (it may be a good description of their precursor based on their initial film characterization, but it is not a good description of their active Cu/CuO material upon reduction).

Instead, a better description would be a nano-structured Cu/CuO material synthesized through electrochemical deposition/decomposition of a molecular precursor. This type of synthesis is somewhat prevalent especially in the water splitting literature (Schley, N. D.; Blakemore, J. D.; Subbaiyan, N.

K.; Incarvito, C. D.; D'Souza, F.; Crabtree, R. H.; Brudvig, G. W. *J. Am. Chem. Soc.* 2011, 133, 10473; Stracke, J. J.; Finke, R. G. *J. Am. Chem. Soc.* 2011, 133, 14872.; Cobo, S.; Heidkamp, J.; Jacques, P.-A.; Fize, J.; Fourmond, V.; Guetaz, L.; Jusselme, B.; Ivanova, V.; Dau, H.; Palacin, S.; Fontecave, M.; Artero, V. *Nat. Mater.* 2012, 11, 802.; Kaeffer, N.; Morozan, A.; Fize, J.; Martinez, E.; Guetaz, L.; Artero, V. *ACS Catal.* 2016, 6, 3727-3737; and many more). In fact, it is often an inadvertent byproduct of attempting to do electrochemistry with a molecular catalyst under aqueous conditions!

This approach is also known in the CO₂ reduction literature contrary to the authors' assertion (see: Huan, T. N.; Andreiadis, E. S.; Heidkamp, J.; Simon, P.; Derat, E.; Cobo, S.; Royal, G.; Bergmann, A.; Strasser, P.; Dau, H.; Artero, V.; Fontecave, M. *J. Mater. Chem. A* 2015, 3, 3901-3907.; Han, Z.; Kortlever, R.; Chen, H.-Y.; Peters, J. C.; Agapie, T. *ACS Central Science* 2017, 3, 853-859 [ref 50], and Hoang, T. T. H.; Ma, S.; Gold, J. I.; Kenis, P. J. A.; Gewirth, A. A. *ACS Catal.* 2017, 7, 3313-3321 [ref 51]). An electrochemist reading this work would say that the three refs above and this work are all examples of heterogeneous Cu materials produced from electrosynthetic degradation of molecular precursors. Instead of burying these works early in introduction, the authors should explicitly contrast their work to these 3 previous examples in the results and discussion to aid readers in understanding why their work is different.

2. Along the lines of putting their work into context, the authors should explicitly compare their results to the other recent manuscript which formed dendritic copper surfaces via a different means that showed high activity for CO₂ reduction to ethylene, ethanol, and 1-propanol (see: Rahaman, M.; Dutta, A.; Zanetti, A.; Broekmann, P. *ACS Catal.* 2017, 7, 7946-7956.)

3. The authors argue that the structured materials afford longer residence time of the intermediates which leads to C-C coupling. This may be true, but could it also be an effect of pH gradients in the material due to increased H⁺ consumption/OH⁻ generation. There are numerous excellent studies looking at the effects of local pH (and relatedly buffer capacity) on CO₂ product distribution at metal catalysts (studies by M.T.M. Koper, Y. Surendranath, M. Gattrell, P. Strasser, and many others). The authors should consider whether the high surface area of their materials leads to increased pH which, in turn, leads to higher C₂ production. You have no buffer present, so you should have dramatic swings in local pH! The authors must discuss this increasingly important aspect of CO₂RR as they discuss the data.

4. The introduction focuses a great deal on molecular catalysts for CO₂RR. While I understand the authors desire for this given their molecular precursors, I do not understand why this is relevant anymore to this manuscript given that the active species is a Cu/CuO dendritic structure resulting from reductive degradation of their in situ generated electrodeposited putative molecular assemblies. I think a better use of the introduction would be to cut back on the molecular catalyst section and instead focus on the use of molecular complexes for electrosynthetic growth of active catalyst films (as highlighted in comment 1). The authors could then free up space and references to talk more about the mechanism, and the possible role of pH in the mechanism, in the discussion portion of the manuscript.

Reviewer #3 (Remarks to the Author):

The authors have made changes based on the reviewers' comments. The current manuscript is better in quality. However, there are still some issues should be addressed before publication.

1. The authors still haven't provided evidence to answer why complex-1 presents the best activity, and what kind of role does the ligand play during CO₂ reduction.
2. In situ spectroscopic techniques are recommended to identify the possible intermediates and elucidate the mechanism for CO₂ reduction.

3. Why KCl is selected as electrolyte? This is not a common electrolyte for CO₂ reduction. As far as I know, Cl⁻ can be electrochemical reduced at the potential range applied in this work.

Responses to the comments and revisions made

Referee 1

Remarks to the Author

This manuscript is of very high quality. The authors have done a substantial revision to the previously submitted version, which I highly appreciate. Although some of the earlier conclusions/claims have been modified/updated, the novelty and significance of this work still stands. It is particularly important that the authors have figured out the real catalytic sites via meticulous characterization. Controlled experiments were also carried out to interrogate the reaction mechanism. All my concerns for the previous version have been properly addressed. I recommend acceptance.

Response: We thank the reviewer very much for the comment.

Two minor points:

1. While reading the experimental procedure, it was at least vague to me how the electrode was protected from being exposed to air during the transfer from the electrochemical cell to the XPS chamber. It is known that reduced Cu surface is sensitive to air (especially when the particle size is small) and that oxidized Cu species play important roles in the catalysis of CO₂ reduction. I thus suggest the authors elaborate on that. See for example: *J. Phys. Chem. C* 2018, 122, 2848-2853.

Response: We thank the reviewer again for the comment. We agree with the referee that the Cu species is sensitive to air, especially the reduced Cu species formed during electrolysis. Therefore, all the samples were immediately immersed in Ar-prepurged acetone and prepared in Ar-filled glove box to protect the catalyst underneath from directly contacting the air before the XPS measurement. The protective sample transfer procedure was similar to those reported in the literature (Ref. Wu Z. S. et al. *J. Phys. Chem. C* 2018, **122**, 2848-2853). According to the comment, in the revised manuscript we have cited the reference (Ref. 13 in supplementary information). We have also emphasized this by “The protective sample transfer procedure was similar to that

reported in the literature¹³. For investigating the evolution of Cu species in the reaction process, catalysts were electrolyzed with different times in the CO₂-saturated electrolytes. After that, the samples were immediately immersed in Ar-prepurged acetone and then transferred into an Ar-filled glove box to protect the catalyst from direct contacting of air. Then, the sample was cleaned, dried and cut into 3×3 mm and glued on a stage with a double sided adhesive. The stage was evacuated into vacuum to prevent the sample to be oxidized in the air. The subsequent testing processes were the same as that of the common X-ray photoelectron spectroscopy.” (Please see Page 7 in the supplementary information of the revised manuscript). We believe that after adding the information, the experimental set-up and procedures are clear now.

2. Judging from the potential-dependent product selectivity, acetic acid and ethanol seem to be competing for electrons. Is it possible that acetic acid is an intermediate for ethanol formation? Can the catalyst directly reduce acetic acid?

Response: We thank the reviewer again for the comment. We agree with the referee that acetic acid and ethanol may be competing for electrons in the reaction. To investigate whether acetic acid was an intermediate for the formation of ethanol or not, some new tests were carried out by replacing CO₂ with acetic acid. However, performing the same experiment with acetate did not yield detectable reduction products, and the result is provided in Table S14 in the revised manuscript. The result indicated that acetic acid was not the intermediate for the formation of ethanol. This result also suggests that acetate cannot be reduced further under our experimental condition, which is in agreement with the conclusion of the previous study (Bertheussen, E. et al. *Angew. Chem. Int. Ed.* 2016, **55**, 1450-1454). In the revised manuscript, we cited this paper (Ref. 109 in the supplementary information). We have also discussed this by “Moreover, adding of acetate did not yield detectable reduction product, this result suggests that acetate cannot be reduced further under our experimental condition, which is in agreement with the conclusion of the previous study.¹⁰⁹” (Please see Page 71 and 74 in the supplementary information of the revised manuscript).

Referee 2

Remarks to the Author

The authors have dramatically revised their manuscript to answer reviewer concerns. Perhaps most transformative is their more comprehensive characterization of the electroactive material from the electrodeposited film which they now suggest is a Cu/CuO dendritic structure. They have attempted to characterize what they consider a “molecular” film via spectroscopic techniques, and they then electrochemically degrade this to a Cu/CuO by applying a sufficiently negative potential. The resulting active CO₂RR films are ligand and time dependent, suggesting that the ligand-metal structure of the initial electrodeposit helps template the final catalytically-active structure. The authors investigate this further for CO₂RR and show good activity for C₂ product formation.

I support publication of this manuscript in Nature Communications based on the characterization of their active material as a structured Cu catalyst and their empirical results involving C₂ product formation. However, I feel that some of the introduction is no longer relevant for their studies, and that the authors neglected to mention the effect of local pH gradients in their discussion of catalytic mechanism—this is a very important discussion in the CO₂ reduction literature on heterogeneous nanostructured materials! The authors should address the following comments prior to publication:

Response: We thank the reviewer for the comment. .

1. The authors should do a much broader literature search of the electrochemical decomposition of molecular precursors for structured electrocatalytic films to better put their system into the broader context of the most relevant literature. For instance, in lines 78-83 they state this is the first example of “conversion of CO₂ using metal organic complex electrocatalysts fabricated by electrosynthesis method.” They then reference a MOF structure as a comparable structure used in sensors and electrical devices. A MOF is not the best description of their system based on their characterization (it may be a good description of their precursor

based on their initial film characterization, but it is not a good description of their active Cu/CuO material upon reduction).

Instead, a better description would be a nano-structured Cu/CuO material synthesized through electrochemical deposition/decomposition of a molecular precursor. This type of synthesis is somewhat prevalent especially in the water splitting literature (Schley, N. D.; Blakemore, J. D.; Subbaiyan, N. K.; Incarvito, C. D.; D'Souza, F.; Crabtree, R. H.; Brudvig, G. W. *J. Am. Chem. Soc.* 2011, 133, 10473; Stracke, J. J.; Finke, R. G. *J. Am. Chem. Soc.* 2011, 133, 14872.; Cobo, S.; Heidkamp, J.; Jacques, P.-A.; Fize, J.; Fourmond, V.; Guetaz, L.; Joussetme, B.; Ivanova, V.; Dau, H.; Palacin, S.; Fontecave, M.; Artero, V. *Nat. Mater.* 2012, 11, 802.; Kaeffer, N.; Morozan, A.; Fize, J.; Martinez, E.; Guetaz, L.; Artero, V. *ACS Catal.* 2016, 6, 3727-3737; and many more). In fact, it is often an inadvertent byproduct of attempting to do electrochemistry with a molecular catalyst under aqueous conditions!

This approach is also known in the CO₂ reduction literature contrary to the authors' assertion (see: Huan, T. N.; Andreiadis, E. S.; Heidkamp, J.; Simon, P.; Derat, E.; Cobo, S.; Royal, G.; Bergmann, A.; Strasser, P.; Dau, H.; Artero, V.; Fontecave, M. *J. Mater. Chem. A* 2015, 3, 3901-3907.; Han, Z.; Kortlever, R.; Chen, H.-Y.; Peters, J. C.; Agapie, T. *ACS Central Science* 2017, 3, 853-859 [ref 50], and Hoang, T. T. H.; Ma, S.; Gold, J. I.; Kenis, P. J. A.; Gewirth, A. A. *ACS Catal.* 2017, 7, 3313-3321 [ref 51]). An electrochemist reading this work would say that the three refs above and this work are all examples of heterogeneous Cu materials produced from electrosynthetic degradation of molecular precursors. Instead of burying these works early in introduction, the authors should explicitly contrast their work to these 3 previous examples in the results and discussion to aid readers in understanding why their work is different.

Response: We thank the reviewer again for the comment. On the basis of the comment, we have made the following modifications.

- 1) After reading the comment, we removed the detailed introduction on

electrocatalytic conversion of CO₂ by complexes. As suggested by the referee, we have discussed more about metal-based catalysts derived from molecular complexes and their application in CO₂ reduction. In the introduction part of the revised manuscript, we have discussed this by adding “Designing robust electrocatalysts is very important for efficient CO₂ reduction^{28,29}. Previous report indicated that heterogeneous metal-based catalysts derived from molecular complexes can be favourable electrocatalysts^{30,31}. The catalytic properties of the catalysts can be tailored by adjusting and regulating the composition of the complexes, and they exhibit excellent catalytic activity toward water splitting and oxygen reduction³²⁻³⁶. It is also an approach towards the fabrication of electrocatalysts, such as additive controlled electrodeposited Cu catalysts for CO₂ reduction³⁷⁻³⁹. However, as a precursor, the complex was generally prepared ex situ or used as a homogeneous additive in decomposition of the new structure in CO₂ electroreduction.”. The papers mentioned by the referee have also been cited (Ref. 32-35 in the revised manuscript). Please see Page 2 in the revised manuscript.

2) As suggested by the referee, we have also discussed the advantage of our catalysts by adding “The immobilization of molecular catalysts onto surfaces provides some obvious advantages over heterogeneous catalysts, such as controllable 3D structure, fast electron transfer rate, etc. “ and “In this work, we turned towards a more direct and facile method to electrodeposit high-surface area Cu-complex film onto conductive substrates. A 3D dendritic Cu-Cu₂O composite catalyst (denoted as Cu-Cu₂O/Cu) was obtained via in situ growth and decomposition of the corresponding Cu-complex film. By using the Cu-complex electrodes as precursor and template, the Cu-Cu₂O/Cu electrode had unique characteristics, such as dendritic 3D structure, near zero contact resistance between the catalyst and the substrate. The in situ synthesized electrode had outstanding catalytic performance for CO₂ reduction to C₂ product in KCl aqueous electrolyte.” Please see Page 2 in the revised manuscript.

3) The papers mentioned by the referee has been cited (Ref. 37-39 in the revised manuscript) and discussed in the results and discussion section. We discussed this by “Using molecular complex as an additive is another method to promote CO₂ reduction

to C₂ product. However, as a precursor, the complex was generally prepared ex situ and used as a homogeneous additive in preparation of the new structure³⁷⁻³⁹. Compared with these methods, the Cu-Cu₂O catalyst synthesized in this work through in situ deposition/decomposition method exhibited controllable 3D structure, high charge transfer rate, and lower overpotential towards liquid C₂ products.” Please see Page 6 in the revised manuscript.

2. Along the lines of putting their work into context, the authors should explicitly compare their results to the other recent manuscript which formed dendritic copper surfaces via a different means that showed high activity for CO₂ reduction to ethylene, ethanol, and 1-propanol (see: Rahaman, M.; Dutta, A.; Zanetti, A.; Broekmann, P. ACS Catal. 2017, 7, 7946-7956.)

Response: We thank the reviewer again for the comment. According to the reviewer’s suggestion, we have discussed this by “Table S8 presents the recent advances in reduction of CO₂ to C₂₊ products. Previous report indicates that dendritic Cu catalysts are capable of reducing CO₂ to C₂ products. These desired structures can be synthesized through different methods. However, the characteristics of the catalyst such as structure and oxidation state are difficult to be controlled, which lead to high overpotential of the C₂ products.” and “Compared with these methods, the Cu-Cu₂O catalyst synthesized through in situ deposition/decomposition method exhibited controllable 3D structure, high charge transfer rate, and lower overpotential towards liquid C₂ products.” Please see Page 6 in the revised manuscript. The relevant references are also included in this section (Ref. 47, 66 and 67 in the revised manuscript) and the literature review in Table S8 of the revised manuscript.

3. The authors argue that the structured materials afford longer residence time of the intermediates which leads to C-C coupling. This may be true, but could it also be an effect of pH gradients in the material due to increased H⁺ consumption/OH⁻ generation. There are numerous excellent studies looking at the effects of local pH (and relatedly buffer capacity) on CO₂ product distribution at metal catalysts

(studies by M.T.M. Koper, Y. Surendranath, M. Gattrell, P. Strasser, and many others). The authors should consider whether the high surface area of their materials leads to increased pH which, in turn, leads to higher C₂ production. You have no buffer present, so you should have dramatic swings in local pH! The authors must discuss this increasingly important aspect of CO₂RR as they discuss the data.

Response: We thank the reviewer for the very instructive comment. We agree with the referee that local pH affects the CO₂ reduction. The rising of local pH in KCl solution may be favorable for C₂ product formation. These are discussed in the following.

1) After reading the comment, we have read more papers about the local pH effect in the CO₂ reduction reaction. Previous study indicated that the local pH on electrode surfaces can influence the CO₂ reduction significantly. It was also revealed that the high local pH induced by the production of hydroxide from the reaction beneficially suppresses the evolution of hydrogen and enhances the selectivity to C₂ products (Hori, Y. In *Modern Aspects of Electrochemistry*, Vayenas, C. G., White, R. E., Gamboa-Aldeco, M. E., Eds. Springer New York: New York, NY, 2008; Vol. 42, pp 89-189; D. R. Raciti, M. Mao, J. H. Park, C. Wang, *J. Electrochem. Soc.* 2018, 165, F799-F804). In KCl solution, the pH will consequently rise in the neighborhood of the electrode in non-equilibrium situation and favor the formation of multi-carbon product. Recent studies have also reported that in KCl solution, C₂ product is preferred to be formed (Zhou Y. S. et al. *Nat. Chem.* 2018, **10**, 974; Lee, S. et al. *Angew. Chem. Int. Ed.* 2015, **54**, 14701-14705; Nakata. K. et al. *Angew. Chem. Int. Ed.* 2014, **53**, 871-874; Yang, H. P. et al. *Green Chem.* 2016, **18**, 3216-3220; Huang, Y. et al. *ChemSusChem* 2018, **11**, 3299-3306). However, in lower local pH, such as concentrated KHCO₃ solution, sufficient HCO₃⁻ is present to neutralize OH⁻. Therefore, the formation of H₂ is favored and C₂ product is less preferred (Y. Hori, A. Murata, R. Takahashi, *J. Chem. Soc., Faraday Trans.* 1989, 85, 2309-2326).

2) To investigate whether the local pH affects the product selectivity significantly, we have also carried out experiments using KHCO₃ solutions. The result indicates that

the current density is higher in KCl electrolyte at all applied potentials (Fig. S30). The maximum FE occurred at -0.4V vs RHE in KHCO₃ aqueous electrolyte, where the FE of ethanol and formic acid (instead of acetic acid) were 47.6% and 32.4%, respectively (Figs. S31, S32 and Table S11). Compared with KCl electrolyte, C₂ product selectivity in KHCO₃ electrolyte was much lower and C₁ product such as formic acid was mainly formed. This is consistent with the above conclusion.

The above results and discussion are given in Figs. S30-32 and Table S11, Please see Page 52-54 in the supplementary information.

In the revised manuscript, we have added some discussion on local pH by “Electrolyte also plays an important role in the CO₂ reduction. Recently, researchers have reported the reduction of CO₂ to C₂₊ products in KCl aqueous electrolyte^{8,68-71}, and it was found to decrease the overpotential and increase the C₂ selectivity. The rise in local pH facilitated a higher amount of adsorbed *CO, which promotes their C-C coupling to C₂ products^{70,72-77}. To investigate whether the local pH affected the product selectivity, we have also carried out experiments using KHCO₃ electrolyte (Figs. S30-S32 and Table S11). The linear sweep voltammograms using CO₂ saturated KHCO₃ aqueous electrolytes of different concentrations (Fig. S31a) illustrate that maximum current density was obtained in 0.5 M KHCO₃ solution. The electrolysis experiments showed that the current densities and selectivity to C₂ products in KCl electrolyte were higher than those in KHCO₃ electrolyte at all applied potentials (Figs. S31c), indicating that the electrolytes influence the electrochemical reaction significantly. One of the main reasons may be that in concentrated KHCO₃ solution, sufficient HCO₃⁻ is present to neutralize OH⁻. Therefore, the formation of H₂ is favored and C₂ product is less preferred⁷². The result suggests that local pH influenced the selectivity of the reaction, which is consistent with results in the recent studies^{8,67-71}.” Please see Page 6 and Page 7 in the revised manuscript.

4. The introduction focuses a great deal on molecular catalysts for CO₂RR. While I understand the authors desire for this given their molecular precursors, I do not understand why this is relevant anymore to this manuscript given that the active

species is a Cu/CuO dendritic structure resulting from reductive degradation of their in situ generated electrodeposited putative molecular assemblies. I think a better use of the introduction would be to cut back on the molecular catalyst section and instead focus on the use of molecular complexes for electrosynthetic growth of active catalyst films (as highlighted in comment 1). The authors could then free up space and references to talk more about the mechanism, and the possible role of pH in the mechanism, in the discussion portion of the manuscript.

Response: We thank the reviewer again for the comment. According to the comment, we have shortened the description about the molecular complexes, and talk more about the mechanism as discussed above.

Referee 3

Remarks to the Author

The authors have made changes based on the reviewers' comments. The current manuscript is better in quality. However, there are still some issues should be addressed before publication.

Response: We thank the reviewer for the comment.

1. The authors still haven't provided evidence to answer why complex-1 presents the best activity, and what kind of role does the ligand play during CO₂ reduction.

Response: We thank the reviewer again for the comment. The comment can be divided into two interesting questions, which are discussed separately below.

- 1) Why complex-1 presents the best activity.

As suggested by the referee, we tried to find the reason for the difference when different complexes were used. The possible reason may be that the intrinsic properties of the complex affect the formation of Cu-Cu₂O catalyst. First, Complex-1 has the lowest contact resistance with the Cu substrate among all the complexes. It leads to the lowest contacting resistance with the Cu substrate among the Cu-Cu₂O in situ formed,

which is contributed to fast electron transfer rate. Based on the comment, we carried out new experiment to measure the charge transfer resistance of the complex before and after electrolysis (Table S13B), and the results are consistent with the conclusion. On the other hand, Complex-1 has the largest surface roughness factor (Figure S11, Table S4), which was favorable to generation of Cu-Cu₂O with abundant exposed active sites, and thus the catalytic activity was higher. We have discussed this by “Among all the electrodes, Cu-Cu₂O-1/Cu has the best activity, which may results from the lowest charge transfer resistance between electrolyte and electrode (Table S13B). In addition, the largest surface roughness factor of the Complex-1 precursor (Fig. S11, Table S4) favored the generation of Cu-Cu₂O with abundant exposed active sites. All these factors can enhance the catalytic activity of the electrode.”. Please see Page 9 in the revised manuscript.

The above results are given in Fig. S11, Table S4 and Table S13B, Please see Page 23, 25 and 66 in the supplementary information.

2) What kind of role does the ligand play during CO₂ reduction.

We think the presence of ligand is versatile for the fabrication of precursor, which is important for the formation of the dendritic Cu-Cu₂O with 3D structure and abundant exposed active sites. In order to study the function of the ligands, we designed some control experiments. In these experiments, we conducted the reaction using metallic Cu foil, Cu foam Cu₂O, and CuO that obtained by direct electrodeposition without using ligand, The SEM results show that dendritic Cu-Cu₂O structure can be formed in the presence of the ligand (Fig. S19), but the structure is unable to form in the absence of the ligand (Fig. S27). The electrolysis results also showed that FE of the C₂ product over Cu foil, Cu foam, Cu₂O, and CuO were all much lower than that over the complex derived Cu-Cu₂O (Fig. 3). It indicates that the ligand is crucial for the formation of Cu-Cu₂O structure, which leads to high FE of the C₂ products. We have discussed this by “Third, the ligand in the complex is crucial for the formation of precursor, which is important for the construction of Cu-Cu₂O structure (Fig. S19). This can be known from the fact that the FE over other Cu-based electrodes, such as metallic Cu,

electrodeposited Cu foam, Cu₂O and CuO was much lower (Fig. 3, Fig. S27). ” and “The ligand is versatile for the fabrication of Cu complexes, which have different morphologies, lattice parameters, and spatial structures. Therefore, the growing interconnected Cu and Cu₂O grains from the constrained environment of the precursor, the obtained dendritic Cu₂O/Cu catalytic electrode surfaces were different apparently in morphologies, electrochemical active areas and the ratio of Cu₂O and Cu on the surface of the catalysts. All of them resulted in the dendritic structure with quantity of the active sites.”. Please see Page 8 in the revised manuscript and Page 29 in the supplementary information.

The above results are given in Fig. 3, Fig. S19 and Fig. S27, Please see Page 7 in the revised manuscript and Page 30, Page 41 in the supplementary information.

2. In situ spectroscopic techniques are recommended to identify the possible intermediates and elucidate the mechanism for CO₂ reduction.

Response: We thank the referee for the comment. We agree with the referee that in-situ IR spectroscopy is a very useful technique to study the intermediate during a reaction. However, we cannot carry out the experiment because the porous electrode which consists of catalyst film and Cu foil, has strong adsorption of IR light. The detection limit afforded by the spectrometer then lead to weak detect signals of the reduced species. We found also that other researchers used bulk metal electrodes for in situ IR in the reaction (Liu, Y. M. et al. *J. Am. Chem. Soc.* 2015, **137**, 11631-11636; Baruch, M. F. et al. *ACS Catal.* 2015, **5**, 3148-3156; Ye, L. T. et al. *Nat. Commun.* 2017, **8**, 14785). Therefore, we regret that we are unable to get this result since the electrochemical system is not up to the experimental conditions and it needs more special electrochemical cell. Thus, according to the comment, we tried our best to get the evidences to study the reaction pathway, which are discussed in the following.

- 1) We studied the composition of the electrolyte samples taking at different electrolysis times by FT-IR, and the results are given in Supplementary Fig. S46A. The formation of COO⁻ can be verified from the peaks at 1352 cm⁻¹ (bridge-bonded) and

1036 cm^{-1} (C-OH). The bands at 1085 cm^{-1} and 881 cm^{-1} are C–O stretch of ethanol. In addition, the peaks 637 cm^{-1} and 431 cm^{-1} in the fingerprint region can be assigned to the characteristic peak of ethanol. In the revised manuscript, we have discussed this by “Figure S46 shows the IR spectra of the electrolyte after different electrolysis times. The intensity of acetic acid and ethanol increased with increasing electrolysis time, indicating that the amount of C_2 products generated increased with electrolysis time, suggesting the formation of acetic acid and ethanol from electroreduction of CO_2 .”. Please see Fig. S46A in the supplementary information and Page 10 in the revised manuscript.

2) We would like to mention that in the previous manuscript that we conducted some control experiments in the presence of the possible reaction intermediates, such as CO, formic acid, formaldehyde, acetaldehyde and acetic acid (Table S14 in Supplementary information) using NMR. From the production rates of acetic acid and ethanol, it can be seen that CO, formaldehyde and acetaldehyde clearly promoted the formation of ethanol. Besides, CO and formaldehyde clearly promoted the formation of acetic acid. Thus, they are reasonable intermediates in the reaction pathway. (Please see Table S14 in the supplementary information).

3) According to the comment of the referee, we also conducted the FT-IR study in the presence of possible reaction intermediates such as CO, formic acid, formaldehyde and acetaldehyde (Fig. S46B). The result indicates that CO and formaldehyde could increase the formation rates of the C_2 products in the electrolysis of CO_2 , but acetaldehyde could only enhance the generation of ethanol. Therefore, we think CO, formaldehyde and acetaldehyde are the possible intermediate in the mechanistic pathway. This conclusion is consistent with the NMR result in Table S14.

We discussed this in supplementary information by “We also conducted the IR study in the presence of possible reaction intermediates. The result indicates that CO, formaldehyde and acetaldehyde are the possible intermediate in the mechanistic pathway. This conclusion is consistent with the result in Table S14.” Please see Page 10 in the revised manuscript and Fig. S46B in the supplementary information.

The above results are given in Table S14 and Fig. S46, Please see Page 73 and 74 in the supplementary information.

3. Why KCl is selected as electrolyte? This is not a common electrolyte for CO₂ reduction. As far as I know, Cl⁻ can be electrochemically reduced at the potential range applied in this work.

Response: We thank the reviewer again for the comment. After reading the comment, we have determined the gaseous products at the potential range applied in our work, and no Cl₂ was detected. This indicates that KCl is stable at the applied potential. Recently, many papers have reported the reduction of CO₂ in KCl aqueous electrolyte, and C₂ product was produced (Zhou Y. S. et al. *Nat. Chem.* 2018, **10**, 974; Lee, S. et al. *Angew, Chem. Int. Ed.* 2015, **54**, 14701-14705; Nakata, K. et al. *Angew. Chem. Int. Ed.* 2014, **53**, 871-874; Yang, H. P. et al. *Green Chem.* 2016, **18**, 3216-3220; Huang, Y. et al. *ChemSusChem* 2018, **11**, 3299-3306). These results also indicated that KCl aqueous is capable to be used even at more negative potential. It was also reported that the adsorbed anions could decrease the overpotential and increase the CO₂ electroreduction rate on Cu based electrode without losing their intrinsically high C₂ selectivity. (Huang, Y. et al. *ChemSusChem* 2018, 11, 3299-3306; McCrum, I. T. et al. *Electrochimica Acta* 2015, 173, 302-309; Varela, A. S. et al. *ACS Catal.* 2016, 6, 2136-2144; Gao, D. et al. *ACS Catal.* 2017, 7, 5112-5120; Gao, D. et al. *Nat. Catal.* 2019, 1).

As suggested by the reviewer, we emphasized this by “Electrolyte also plays an important role in the CO₂ reduction. Recently, many researchers have reported the reduction of CO₂ to C₂₊ products in KCl aqueous electrolyte^{8,68-71}, and it was found to decrease the overpotential and increase the C₂ selectivity. The rise in local pH facilitated a higher amount of adsorbed *CO, which promotes their C-C coupling to C₂ products^{70,72-77}.”. Please see page 6 in the revised manuscript.

Reviewer #1 (Remarks to the Author):

OK to publish

Reviewer #2 (Remarks to the Author):

The authors have sufficiently addressed my concerns. They have restructured the narrative of their introduction to better focus on their studies, and they have changed the discussion of their material's activity for CO₂ reduction to C₂ products to better compare it to similar systems in the field. They have also sufficiently acknowledged the role of local pH in their product distribution. Based on the extensive characterization of their active material as a structured Cu catalyst for CO₂ reduction to C₂ products reported in the study, I believe this manuscript has sufficient novelty and impact for Nature Communications and I recommend its publication without further revision.

Reviewer #3 (Remarks to the Author):

The authors have revised the manuscript based on the reviewers' comments. All my concerns have been properly addressed. I recommend the acceptance of this manuscript for publication in Nature Communications.